# Conservation of Green and White Ash Germplasm Using the Cryopreservation of Embryogenic Cultures

**DOI:** 10.3390/plants13030352

**Published:** 2024-01-24

**Authors:** Mason Richins, Cristian Montes, Scott Merkle

**Affiliations:** Warnell School of Forestry and Natural Resources, University of Georgia, Athens, GA 30602, USA; mwr79514@uga.edu (M.R.); cristian.montes@rayonier.com (C.M.)

**Keywords:** *Fraxinus americana*, *Fraxinus pennsylvanica*, cryostorage, somatic embryogenesis, emerald ash borer, in vitro culture, vitrification

## Abstract

Green ash (*Fraxinus pennsylvanica*) and white ash (*F. americana*) populations are currently experiencing major declines across their native ranges in North America due to infestation by the exotic insect pest emerald ash borer (*Agrilus planipennis*). The development of a reliable method for the long-term storage of green and white ash germplasm in the form of embryogenic cultures using cryopreservation would be a considerable aid to ash conservation efforts. We compared recovery percentages of cryopreserved green and white ash embryogenic cultures using vitrification versus slow cooling methods. Three Plant Vitrification Solution 2 (PVS2) exposure durations (40, 60, and 80 min) for vitrification and three DMSO concentrations (5%, 10%, and 15%) for slow cooling were tested for their effects on the percentage of cultures that regrew following cryostorage. Vitrification resulted in a higher overall culture recovery percentage (91%) compared to cultures that were cryostored using the slow cooling approach (39%), and a more rapid initiation of regrowth (5 days versus 2–3 weeks) resulted. Recovery from cryostorage by cultures using the slow cooling approach varied significantly (*p* < 0.05) between experiments and with genotype (*p* < 0.05). The recovery of vitrified tissue from cryostorage did not vary with genotype, species, or PVS2 exposure duration (*p* > 0.05). The vitrification cryopreservation protocol provides a reliable and versatile alternative to the traditional slow cooling method, strengthening our ability to preserve valuable ash germplasm for conservation and restoration.

## 1. Introduction

The ash genus (*Fraxinus*) contains up to 25 species native to North America [1]. Two ash species, green ash (*Fraxinus pennsylvanica*) and white ash (*Fraxinus americana*), are important trees in much of the eastern and midwestern United States. Ash trees are a major food source and provide shelter for numerous native terrestrial and aquatic wildlife species, while ash wood is highly valued commercially for its strength, hardness, and durability, making it an ideal material for baseball bats, tool handles, furniture, and other applications [2,3]. Due to the introduction of the invasive emerald ash borer (*Agrilus planipennis*; EAB), the future of ash trees and their ecological and economic roles are now under threat. EAB is a wood-boring buprestid beetle native to Asia that was introduced to the United States in the early to mid-1990s. Infestations rapidly spread to 35 US states and several Canadian provinces, killing millions of ash trees in the process [4]. Minimal EAB resistance has been identified in native North American ash, although ongoing breeding and genetic transformation programs are seeking to address this [5,6]. Other mitigation strategies, such as long-term biological control using specialized parasitoid wasps, are being developed, but scalability and testing remain challenges [7].

The delivery of pest- and pathogen-resistant genotypes from tree breeding programs is an arduous process that may span multiple decades. As such, the preservation and propagation of valuable germplasm are essential to conservation efforts and may be aided by employing in vitro clonal propagation systems, such as somatic embryogenesis (SE). SE from zygotic embryo explants has been achieved in multiple ash species, including Manchurian ash (*Fraxinus mandshurica*), green ash (*Fraxinus pennsylvanica*), white ash (*Fraxinus americana*), and common ash (*Fraxinus excelsior*) [8,9,10,11,12]. The long-term maintenance of valuable embryogenic material via serial transfer remains problematic, however, as cultures can be lost to contamination and tend to decline in embryogenic potential over time. Long-term preservation via cryopreservation provides an effective solution to these issues.

Cryopreservation via liquid nitrogen (LN; −196 °C) is the primary biotechnology approach utilized for the long-term storage of plant material. Storage at −196 °C halts cellular metabolic activity in the tissue, allowing storage for an indefinite duration with minimal risk to tissue viability [13]. One of the original techniques established for successful cryopreservation is the slow cooling method, also known as the two-step freezing or controlled freezing method. Slow cooling relies on low concentrations of membrane-permeable and -impermeable cryoprotectants, in combination with a gradual, controlled temperature drop [14]. The prevention of intracellular ice formation during freezing and storage is essential for all cryopreservation techniques, and with slow cooling, this is accomplished via cellular dehydration using osmotica and cryoprotective solutions [15]. DMSO, incorporated into the liquid culture medium, is generally used as the slow cooling cryoprotectant and sorbitol or sucrose as the osmoticum. Recovery from slow cooling has been reported to be species- and tissue-specific, with the highest success reported for cell cultures and undifferentiated tissue [14,16,17]. Extracellular freezing does occur with this technique, which reduces success when storing complex tissues. The optimal cooling rate is dependent on the solute concentration and tissue type; however, a cooling rate of −1 °C/min is generally considered ideal to minimize cellular damage from rapid osmotic changes and intracellular ice crystal formation. Tissue samples are cooled at this rate down to −80 °C and then transferred into LN [14]. Success with this approach has been demonstrated using undifferentiated and differentiated tissues across multiple woody plant species [18,19].

An alternative to the slow cooling approach is the cryopreservation technique known as vitrification. Unlike the slow cooling method, vitrification inhibits intra- and extracellular ice formation by employing high concentrations of cryoprotectants and rapid cooling rates (15,000–30,000 °C/min) via direct exposure to LN [20]. The viscosity of the solution and cell cytosol rapidly increases during cooling until it solidifies at approximately −115 °C, entering an amorphic “glass-like” state without ice crystallization [21]. By avoiding ice nucleation and freezing, vitrification is capable of preserving a broader range of complex tissue types and organs with reduced risk of mechanical and osmotic damage [22]. Cryopreservation via vitrification has been successfully employed for multiple woody species and plant organs, including meristems, shoot tips, hairy roots, and somatic embryos [23]. Among forest trees, apical and basal buds from redwood (*Sequoia sempervirens*) in vitro shoot cultures and black alder (*Alnus glutinosa*) and cork oak (*Quercus suber*) embryogenic material have been successfully cryopreserved using vitrification [19,24,25].

The most commonly used cryoprotectant solution for embryogenic tissue storage via vitrification is Plant Vitrification Solution 2 (PVS2), developed by Sakai et al. [21], which contains glycerol, DMSO, and ethylene glycol. The most frequently reported vitrification protocols include two pretreatment phases (preculture and loading), PVS2 exposure, and, for recovery from cryostorage, an unloading phase following removal from LN and rewarming [19]. The preculture, loading, and PVS2 phases enhance dehydration tolerance while gradually dehydrating the cells and increasing intracellular solute concentrations [26]. The inclusion of these steps was important for the recovery of black alder embryos, increasing the recovery percentage to over 90% [19]. However, it should be recognized that optimal vitrification protocols and recovery percentages vary with species and tissue type, as multiple factors, such as organ size, can influence the cooling rates required for vitrification [20].

Some research on the cryopreservation of ash species has been reported. A 2005 study on the cryopreservation of common ash (*Fraxinus excelsior*) stem tips found that recovery was highest when the stem tips were vitrified [27], while a later study using common ash embryogenic callus reported that vitrification led to lower regrowth percentages compared to the slow cooling method [18]. There are no publications reporting the application of either of these cryopreservation methods to green ash or white ash embryogenic cultures.

The objective of this study was to develop a reliable cryopreservation protocol for the long-term storage of green and white ash embryogenic tissue. Slow cooling and vitrification methods were optimized and then compared to determine which method resulted in higher recovery percentages. Different DMSO concentrations were tested with the slow cooling approach, and different PVS2 exposure times were tested within the vitrification approach. Following recovery from cryostorage, culture material was tested to confirm that it retained its capacity to produce somatic embryos.

## 2. Results

### 2.1. Vitrification Experiments

The first evidence of regrowth following cryostorage using vitrification was observed only five days following rewarming, rinsing, and final plating on semisolid IMM for 5-FP-5, LA112-10, and LA115-5. Proembryogenic mass (PEM) regrowth generally developed as single events along the PEM clusters, with the white to yellow color of the new growth contrasting with the dark brown of the unrecovered tissue (Figure 1b). Interestingly, some genotypes (e.g., LA115-5) consistently regrew from multiple points simultaneously over much of the thawed tissue clusters (Figure 1c). Regrown PEM clusters resumed regular embryogenic appearance, with similar growth and vigor to that of the original cultures maintained on semi-solid IMM prior to cryostorage. Non-morphogenic callus composed of vacuolated cells was also observed, although at a lower frequency than embryogenic tissue, originating primarily from developing somatic embryos rather than PEMs. The total culture recovery percentage remained at or above 80% for each of the three vitrification experiments. No significant differences (*p* > 0.05) were found among the three experiments when experiment 1 was used as the baseline comparison, so data were combined from all three experiments to analyze genotypic and treatment effects. An overall recovery percentage of 89.3% was achieved for all vitrification treatments and genotypes (5-FP-4, 5-FP-5, LA112-10, and LA115-5). Despite the variation in culture line growth rate and appearance prior to cryopreservation, recovery across the three experiments was above 70% for each genotype and did not vary significantly (*p* > 0.05) among genotypes when compared with 5-FP-4 as the model baseline. Genotypes LA112-10, LA115-5, and 5-FP-4 had overall recovery percentages of 90% or above, while 5-FP-5 had an overall recovery percentage of 71%, with the lowest percentage of 67% for the 40 min of PVS2 exposure treatment (Figure 2).

All cultures exposed to pretreatments (preculture and loading) and PVS2 for 40, 60, or 80 min at 0 °C in liquid IMM without cryostorage resumed the proliferation of PEMs following transfer to semi-solid IMM (Figure 2) and appeared similar to material that had been cultured in liquid IMM with no pretreatments or PVS2, followed by transfer to semisolid IMM. The effect of the three PVS2 exposure times (40, 60, and 80 min) on the regrowth percentage of the cryostored cultures and non-cryostored cultures did not vary significantly (*p* > 0.05) when the 60 min treatment was used as the model baseline, nor did the appearance of the regrown material differ among the PVS2 exposure times. No significant interactions (*p* > 0.05) were found between genotype and PVS2 exposure time.

Bacterial contamination was a persistent problem in all three experiments. Over 20 of the 180 plates across the three experiments were observed with contamination prior to embryogenic culture regrowth. Two plates from experiment 1 and one plate from experiment 2 developed vigorous contamination overgrowth following rewarming and were omitted from the final analysis. Despite this issue, in most contaminated plates, embryogenic regrowth could be observed and recorded before contamination overgrew the Petri dishes, so these plates were included in the analysis. The water in the hot water bath during rewarming was suspected to be the primary source of contamination in experiments 1 and 2. Experiment 3 cryovials were rewarmed in Milli-Q water that was isolated from the water in the main hot water bath, resulting in a substantial reduction in contamination percentages in the rewarmed cultures.

### 2.2. Slow Cooling Experiments

Although the slow cooling experiment was repeated three times, none of the cryostored samples from any of the treatments or controls recovered in the third experiment, and this was probably due to an unknown error in the handling of the samples. Therefore, data from that experiment were not included in the analysis. In contrast to the vitrification experiment, regrowth following cryostorage using slow cooling varied significantly between the two experiments that were used in the data analysis, so the experiment remained as a factor in the analysis of this experiment (i.e., data from the two experiments were not combined for GLM analysis). The overall regrowth percentages in the two experiments were 42% and 36%, both of which were much lower than those of the vitrification experiment. Additionally, the first signs of regrowth appeared 2–3 weeks following thawing, which was a much longer delay than that which followed rewarming in the vitrification experiments. Unlike vitrification, recovery from cryostorage with slow cooling was genotype-dependent (Figure 3). Genotypes 5-FP-4 and LA112-10 both had significantly lower regrowth percentages (*p* < 0.01) when compared with LA115-5 as a model baseline. LA115-5 also regrew faster than 5-FP-4 or LA112-10, with an abundance of simultaneous sites of regrowth among the PEMs (Figure 1d). The recovery of 5-FP-4 had the lowest regrowth percentage, and it generally regrew the slowest. The contrast in performance among different genotypes was particularly striking in the second slow cooling experiment, in which genotype 5-FP-4 showed no recovery from cryostorage with any pretreatment, while 93% of LA115-5 samples from all DMSO treatments recovered.

As with the vitrification experiment, all cultures exposed to slow cooling pretreatments (0.4 M sorbitol and 5%, 10% or 15% DMSO in liquid IMM) without cryostorage resumed the proliferation of PEMs following the transfer to semi-solid IMM (Figure 3). Tissue that did not undergo pretreatments and was cryostored suspended in liquid IMM without cryoprotectants had 0% recovery for 5-FP-4 and LA112-1 and 30% regrowth for LA115-5, compared to 70% to 90% regrowth for the pretreated samples of this culture line (Figure 3). Across all culture lines, cryostorage using the 15% DMSO treatment with 0.4 M sorbitol resulted in 46% recovery, the highest percentage among the tested concentrations. A DMSO concentration of 10% (the standard concentration used in our laboratory for other hardwood species) resulted in the lowest percentages among the three DMSO concentrations tested. While these data may appear to indicate that the addition of the cryoprotectant solution with 15% DMSO enhanced the regrowth percentage following cryopreservation compared to other DMSO treatments and samples that were cryostored without pretreatment, the variation between the two slow cooling experiments was very large, as indicated by the standard error bars in Figure 3. Thus, GLM did not identify a significant difference among the DMSO treatments when using 10% DMSO as an intercept baseline.

### 2.3. Somatic Embryo Production

Mature embryos were produced from all cryostored and regrown culture lines, with the exception of 5-FP-5. However, only somatic embryos derived from recovered tissue from the vitrification experiments were matured for somatic seedling production due to the high recovery percentage for vitrification as compared to slow cooling. Somatic embryos completed development up to the cotyledonary stage on semi-solid EDM in 1 to 2 months, similarly to embryos from non-cryostored cultures. Matured embryos were healthy and exhibited typical morphological traits (Figure 4).

## 3. Discussion

Somatic embryogenesis provides a promising mass propagation tool for the restoration of threatened forest tree species. However, the long-term maintenance of embryogenic culture lines via serial transfer is labor-intensive, and the cultures are prone to decline with respect to embryogenic potential over extended periods. Germplasm preservation via cryopreservation mitigates these issues and allows the long-term storage of valuable genotypes and the maintenance of biodiversity. In this study, we established a reliable cryopreservation technique for green ash and white ash embryogenic tissue, employing PVS2-based vitrification. Vitrification outperformed the slow cooling approach, allowing the storage of valuable ash germplasm with consistently high overall (>89%) percentages of recovery across all tested genotypes of both species. The vitrification pretreatments and treatments employed also yielded minimal phytotoxicity and no loss of culture lines. In contrast to the results reported here, the vitrification-based cryostorage pretreatment applied to common ash embryogenic tissue failed to yield any recovery of viable calluses following cryostorage, while slow cooling using sucrose as osmoticum and DMSO for cryoprotection was much more successful [18]. Since the vitrification protocol applied to common ash differed from the protocol that we applied to green ash and white ash, it is not possible to say which factors may be responsible for the different results.

Vitrification-based cryopreservation works through the use of high concentrations of cryoprotectant solutions in conjunction with rapid cooling [21]. Tissue exposed to cryoprotectants within PVS2 prior to LN submergence lowers available water within the cells and alters freezing properties to protect the plant tissue from ice-related damage as it enters a glass-like state when cooled to −196 °C [28]. However, the constituents of PVS2, DMSO and ethylene glycol, are toxic to plant tissues, and long durations of exposure to them can severely damage cells. Thus, identifying the optimal exposure time is necessary to maximize the recovery percentages of the material to be cryostored. Previous studies showed that the optimal PVS2 pretreatment duration at 0 °C prior to storage in liquid nitrogen is species- and tissue-dependent [29]. Optimal PVS2 exposure duration is also indirectly influenced by tissue mass, as larger tissue masses require a more rapid decrease in ambient temperature to prevent ice nucleation and growth throughout the tissue. Thus, larger tissue masses may undergo more severe osmotic stress [30]. The small masses of the ash PEMs cryostored in the vitrification experiment were apparently advantageous for survival at all tested PVS2 exposure durations. In the current study, green and white ash embryogenic tissues were found to be tolerant to PVS2 exposures of 40, 60, and 80 min. The highest recovery percentage (93%) was observed with 60 min exposure, although it was not statistically higher than the recovery percentages for the other two durations. This exposure time is consistent with the results reported for the vitrification-treated embryogenic tissue of *Alnus glutinosa*, *Quercus suber*, and *Castanea sativa*, although different optimal exposure durations were reported for other species such as *Aesculus hippocastanum* [31,32,33,34].

The quality of the culture material prior to pretreatment appeared to influence recovery from cryostorage following both vitrification and slow cooling. High-quality tissues were defined as those that grew prolifically while retaining PEM morphology, light yellow to tan tissue coloration, and a clear liquid medium (not clouded by an abundance of vacuolated cells). LA115-5 proliferated the most rapidly in suspension culture and consistently recovered from cryostorage the most rapidly in the vitrification and slow cooling experiments. It is also interesting that LA115-5 was the only tested genotype that recovered from storage in LN without pretreatment, although at a lower percentage than pretreated replicates. Genotype 5-FP-5 displayed the slowest regrowth following recovery from cryostorage and the lowest post-cryostorage recovery percentage (71%) in the vitrification experiments compared to the other genotypes; however, this percentage was still greater than the average of all lines that underwent slow cooling cryostorage. It was later discovered that 5-FP-5 exhibited reduced embryogenic potential at the time of vitrification pretreatment, which may have also impacted recovery. Embryogenic tissue exposure to PVS2, whether followed by storage in LN or not, did not permanently inhibit normal embryogenic growth. The pretreatment resulted in high recovery percentages regardless of species or genotype, indicating that this method may be applicable to embryogenic cultures of a broad range of additional ash genotypes and species.

Unlike vitrification, slow cooling methods require a gradual decrease in temperature at a rate of approximately 1 °C/min in combination with low concentrations of cryoprotectant to minimize osmotic stress to cells during extracellular freezing [14]. DMSO enhances cell membrane permeability and cellular dehydration through the addition of an osmoticum, thus limiting available water within the cytoplasm and subsequent damage from ice crystals [15]. As is the case with vitrification, however, the DMSO used in the slow cooling protocol is toxic to cells, and determining an optimal DMSO concentration is essential to cell survival [35]. In this study, the DMSO pretreatment with 0.4 M sorbitol produced the highest percentage of recovery (46%) across all three tested culture lines among the tested DMSO concentrations. A concentration of 0.4 M sorbitol has performed well in other studies, while the optimal DMSO concentration has varied [36]. In that study, PEMs were suspended in the cryoprotectant solution just prior to freezing in an attempt to minimize phytotoxic effects. However, this precaution may not be necessary, as other cryostorage studies succeeded using slow cooling with an extended cryoprotectant pretreatment [18,34]. For the DMSO concentrations tested in this study, a DMSO phytotoxicity threshold was not reached, as all non-cryostored controls regrew. Thus, extended exposure durations or higher concentrations of DMSO than those tested in our study may yield higher recovery percentages for ash tissues.

The recovery of the slow cooled material differed between the two slow cooling experiments in which the regrowth of cryostored material was observed. The recovery percentage of genotypes 5-FP-4 and LA112-10 in the second slow cooling experiment decreased by over 20% across all DMSO treatments while the recovery percentage of LA115-5 increased by over 30% (Figure 3). As previously mentioned, a third slow cooling experiment was conducted; however, no cryostored cultures were recovered after storage in LN, with or without the preculture and DMSO pretreatments. All controls exposed to pretreatments but not cryostored in the third experiment recovered and resumed PEM growth in a similar manner to the previous non-cryostored control replicates. Interestingly, for one genotype (LA115-5), 30% of the samples that received no pretreatment at all prior to cryostorage regrew. While this percentage was much lower than for the pretreated samples of the same genotype, in our experience, we have not noted the recovery of non-pretreated controls from cryostorage occurring with other embryogenic cultures of ash or other hardwood species. This culture line, which we have maintained in the lab for several years, is exceptionally vigorous and rapidly proliferates as small, dense PEMs.

Material cryostored via vitrification resulted in the reliable regeneration of embryogenic tissue and mature embryos from three of the four tested culture lines (LA115-5, LA112-10, and 5-FP-4). The tissue of the 5-FP-5 culture line recovered from cryostorage was similar in appearance to the non-cryostored source culture but did not produce somatic embryos following transfer to EDM and was eventually determined to no longer be embryogenic. However, the transfer of non-cryostored 5-FP-5 tissue to EDM was not carried out to confirm that embryogenic potential was lost due to cryopreservation treatments. Recent studies using PVS2 for vitrification have not reported deleterious morphogenic or developmental effects. San José et al. [19] did not detect DNA ploidy instability in vitrified *Alnus glutinosa*, nor were other large-scale mutations observed in cryostored (by vitrification) *Quercus robur* embryogenic tissue [33].

## 4. Materials and Methods

### 4.1. Plant Material

Embryogenic green ash and white ash cultures used in cryopreservation experiments were derived from culture initiations performed in 2015 and 2018 using immature zygotic embryos, as described previously [37]. Cultures were grown in 60 mm plastic Petri dishes containing semi-solid induction–maintenance medium (IMM) [37], which was a modified Woody Plant Medium (WPM) [38] with 30 g/L sucrose, 0.5 g/L L-glutamine, and either 2 mg/L 2,4-dichlorophenoxyacetic acid (2,4-D) or 0.2 mg/L picloram, with pH adjusted to 5.7. IMM was sterilized via autoclaving, as were all media used in the study. Embryogenic cultures were maintained in the dark at 24 ± 1 °C and transferred to the fresh medium of the same composition on a monthly basis. Embryogenic culture lines were chosen for cryopreservation experiments based on proliferation rates, with preference being for lines that proliferated as proembryogenic masses (PEMs; Figure 1a) rather than as repetitive embryos. Two green ash genotypes (5-FP-4, initiated and maintained on IMM with picloram, and 5-FP-5, initiated and maintained on IMM with 2,4-D) and two white ash genotypes (LA112-10 and LA115-5, both initiated and maintained in IMM with 2,4-D) were selected for the cryopreservation experiments.

To produce sufficient culture material for each of the experiments, approximately 0.5 g of PEMs from each culture line was inoculated into 125 Erlenmeyer flasks containing 30 mL liquid IMM with the same PGRs used in the semi-solid media for proliferation. Embryogenic suspension cultures were grown on orbital shakers at 100 rpm and maintained in the dark at 24 ± 1 °C for at least 45 days, during which the liquid medium was refreshed every three weeks by pipetting out most of the old medium and adding 25 mL of fresh medium. Non-target tissues, such as developing embryos, were removed from the cultures before the cultures were used for cryopreservation experiments.

### 4.2. Vitrification Experiments

The vitrification protocol employed by San José et al. for *Alnus glutinosa* embryogenic tissue [19] was tested with ash embryogenic cultures. Both green ash genotypes (5-FP-4 and 5-FP-5) and both white ash genotypes (LA112-10 and LA115-5) were tested in this experiment. The protocol included the following stages, with each stage involving the use of a unique solution: preculture solution (PS), loading solution (LS), Plant Vitrification Solution 2 (PVS2), and unloading solution (ULS), the formulations of which are detailed in the protocol descriptions below. PVS2 exposure durations tested prior to storage in liquid nitrogen were 40, 60, and 80 min to determine the exposure duration that resulted in the highest percentage of culture regrowth. The recovery of cultures treated with the PS, LS, and PVS2 sequence without storage in LN was tested in experiments 1 and 2 as controls. In experiment 1 only, the regrowth of suspension culture material plated onto semisolid IMM (without pretreatments or LN) was also tested to confirm culture viability. The recovery of tissue stored in LN without pretreatments was not tested in the vitrification experiments, but this control was included in the slow cooling experiments (see below).

For pretreatment, approximately 20 mL of PEMs from the suspension cultures of each genotype was transferred to 125 mL Erlenmeyer flasks containing 30 mL of PS, which consisted of IMM supplemented with 0.3 M sucrose and PGRs (2 mg/L 2,4-D or 0.2 mg/L picloram), consistent with the IMM variant on which the cultures had been maintained. Approximately 5 mL of tissue from each genotype was reserved and plated directly onto semi-solid IMM as a non-pretreated/non-LN control for recovery. Precultures were agitated on an orbital shaker (100 rpm) for 2 days at 24 ± 1 °C in the dark, similarly to the preculture step used by San José et al. [19]. The PS was then decanted, and 30 mL of LS consisting of IMM supplemented with 0.4 M sucrose and 2 M glycerol was added. Cultures were again agitated in the dark under the same conditions for 20 min and then decanted. Then, approximately 0.4 mL of embryogenic clusters was transferred into sterile 2 mL cryovials (Nalgene). Some tissue remained in suspension flasks for the later testing of pretreatment effects without LN storage as additional controls. Once the tissue was placed in all cryovials, prechilled (4 °C) PVS2 comprising IMM with 0.4 M sucrose and supplemented with 30% glycerol, 15% DMSO, and 15% ethylene glycol (*w*/*v*) was added to each cryovial up to the 2 mL line. The cryovials were placed in plastic cryoboxes and stored on ice (0 °C) for 40, 60, or 80 min; then, they were plunged directly into LN (−196° C). The remaining tissue in the flasks that had been treated with PS and LS was transferred to PVS2 under the same conditions for the same durations and then immediately transferred to semi-solid IMM to test recovery from the pretreatment without exposure to LN.

Following storage in LN for at least 48 h, cryovials were rapidly rewarmed in a 40 °C water bath for 3 min. In the first two vitrification experiments, cryovials held in vial racks were placed directly into hot water baths, whereas cryovials in the third experiment were rewarmed in Milli-Q ultra pure water in a separate vessel placed within the main hot water bath. In a laminar flow hood, PVS2 was decanted, and ULS (1.2 M sucrose in water) was added to each cryovial to purge cryoprotectants from the rewarmed PEMs. PEMs were rinsed and decanted in ULS three times for approximately 3 min each. Then, the final rinse was decanted, and PEMs were collected on 2 cm squares of 30 µm pore size sterile nylon mesh (Lamports Filter Media Inc., Cleveland, OH, USA) overlaid on filter paper (Whatman 1) to remove residual ULS. Finally, the nylon mesh squares along with tissue were transferred to plastic Petri plates (60 mm) containing semi-solid IMM supplemented with PGRs consistent with the original media on which the source cultures had been maintained. Cultures (i.e., the PEM contents of each cryovial) were transferred to fresh media after one day and transferred again after one week to dilute any remaining ULS. Cultures were incubated in the dark at 22 °C. Then, they were monitored for two months, after which the total number of cultures from each treatment by genotype combination that displayed regrowth was recorded.

### 4.3. Slow Cooling Experiments

The protocol tested in this experiment was adapted from a method reported for the cryostoring embryogenic cultures of *Liriodendron tulipifera* and *Liquidambar* spp. [36]. It involved three main stages: preculture, two-step freezing, and thawing. Three of the same ash embryogenic culture lines used in the vitrification experiment (5-FP-4, LA112-10, and LA115-5) were used in this experiment. Line 5-FP-5, which was used in the vitrification experiment, was not used in this experiment due to its relatively slow growth in suspension culture and a decline in its embryogenic potential. Three concentrations (5%, 10%, and 15%) of DMSO were tested for their effects on the regrowth of embryogenic tissue following recovery from cryostorage. Two controls were used: exposure to preculture medium and cryopreservation medium without LN storage and storage in LN with no pretreatment.

Approximately 20 mL of tissue from each culture line was transferred from proliferating suspension cultures in IMM to 125 mL Erlenmeyer flasks containing 30 mL of preculture solution consisting of IMM with the appropriate growth regulators (2,4-D or picloram), 30 g/L sucrose and 0.4 M sorbitol. Cultures were agitated on orbital shakers at 100 rpm for 24 h at 24 ± 1 °C in the dark and then chilled at 4° C for 30 min. The preculture medium was decanted, and approximately 0.4 mL of the chilled tissue was transferred into sterile 2 mL cryovials. Prechilled (4 °C) cryopreservation medium consisting of IMM with 30 g/L sucrose, 0.4 M sorbitol, and DMSO (5%, 10%, or 15%) was then added to each cryovial up to the 2 mL fill line. Vials were capped and placed in pre-chilled (4 °C) Nalgene “Mr. Frosty” containers filled with isopropyl alcohol, which were then stored in a −80 °C freezer. The containers with alcohol provided a cooling rate of approximately −1 °C/min down to −80 °C, minimizing damage to the cells from intracellular ice crystal formation. After 24 h, the cryovials were removed from the freezer, transferred to cryoboxes, and stored in LN.

Following at least 48 h of storage in LN, cryovials were removed from LN and thawed in a 40 °C water bath for 2–3 min [36]. Vial contents were collected on 2 cm squares of 30 µm pore size sterile nylon mesh overlaid on filter paper and left for 3 min to allow the liquid cryopreservation medium to be absorbed by the paper. Nylon squares along with the thawed tissue were transferred onto 60 mm plastic Petri dishes with the same semi-solid IMM on which the cultures originally had been maintained. The nylon mesh squares with tissue were transferred to fresh IMM after one day and again after one week and then monitored for two months until embryogenic tissue regrowth was observed.

### 4.4. Somatic Embryo Production from Recovered Cultures

Cultures from each of the tested genotypes that regrew following recovery from LN in the vitrification experiment were selected to be tested for somatic embryo production. These cultures were maintained on semi-solid IMM with monthly transfer to fresh medium until sufficient material was produced for somatic embryo production. Embryogenic tissue was transferred from IMM to a semi-solid embryo development medium (EDM), which was the same formulation as IMM but without PGRs, to allow somatic embryo development.

### 4.5. Experimental Design and Statistical Analysis

Each vitrification and slow cooling experiment was completely repeated three times, although no treatments from the third slow cooling experiment showed regrowth, so regrowth data from this experiment were not included in the Results. Thus, regrowth data from three separate vitrification experiments and two separate slow cooling experiments were analyzed. For each experiment, 5 cryovials of each genotype × treatment combination were cryostored, recovered from LN, and tested for regrowth, for a total of 226 vitrification cryovials and 210 slow cooling cryovials. As described previously, some controls were only given pretreatments, without storage in LN, to determine the effects of the pretreatments on culture regrowth. One 5 mL sample of each genotype was used for these controls in each experiment.

The effects of the treatments tested in the slow cooling and vitrification experiments on embryogenic tissue recovery frequency were analyzed using generalized linear models (GLM) and the RStudio program for a logit response [39,40]. Models were generated for each of the treatments tested using the two cryopreservation methods, with embryogenic culture regrowth being recorded as binary data (1 for regrowth or 0 for no regrowth). Most generated models for each of the methods were tested as single coefficients due to unbalanced replications within the treatments. Coefficients within each model contrast were assigned baseline levels for controls during analysis. Generally, levels with outcomes closest to the result averages were selected as controls. The significance of treatment effects was first determined within each of the two methods and then a final comparison was conducted for the overall regrowth frequency between the two methods. However, because vitrification and slow cooling were tested in separate experiments, it was not possible to statistically compare the regrowth percentages between the two approaches. The divergent protocols and the number of samples made it impossible to handle two sets of cultures, with each receiving a different treatment, simultaneously. In order to visually display the variation in regrowth among the different treatment × genotype combinations, regrowth data were converted to percentages (rather than 0 s and 1 s) and means for these percentages, with standard errors generated for Figure 2 and Figure 3. These percentage data were not used for statistical analysis.

## 5. Conclusions

This is the first report of a reliable cryopreservation protocol optimized for green ash and white ash embryogenic tissues. Vitrification outperformed the slow cooling method, demonstrating its versatile application for the storage of valuable ash germplasm. We believe that it will be a valuable tool for ash conservation and restoration in the face of the EAB infestation in North America and for future cryopreservation research with other threatened forest tree species.

## Figures and Tables

**Figure 1 plants-13-00352-f001:**
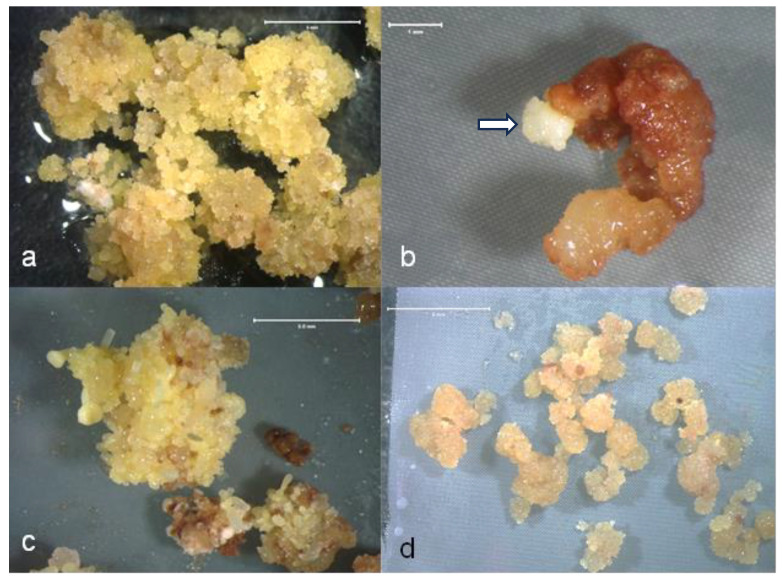
Ash embryogenic culture material used for cryopreservation and regrowth of cultures following cryostorage. (**a**). Green ash PEMs from culture line 5-FP-4 used for cryopreservation. Similar tissue quality was selected for cryostorage from all culture lines for all experiments. Bar = 5 mm. (**b**). Early stage of embryogenic tissue regrowth (arrow) from PEMs of culture line 5-FP-4 following cryostorage using the vitrification method. This culture sample was exposed to PVS2 for 40 min at 0 °C prior to storage in liquid nitrogen. Bar = 1 mm. (**c**). Embryogenic tissue recovery from white ash culture line LA112-16 following cryostorage in the vitrification experiment. This culture sample was exposed to PVS2 for 40 min at 0 °C prior to storage in liquid nitrogen. Bar = 5 mm. (**d**). Embryogenic tissue recovery from white ash culture line LA115-5 following cryostorage using the slow cooling method. This culture was suspended in medium supplemented with 5% DMSO during cryostorage pretreatments. Bar = 5 mm.

**Figure 2 plants-13-00352-f002:**
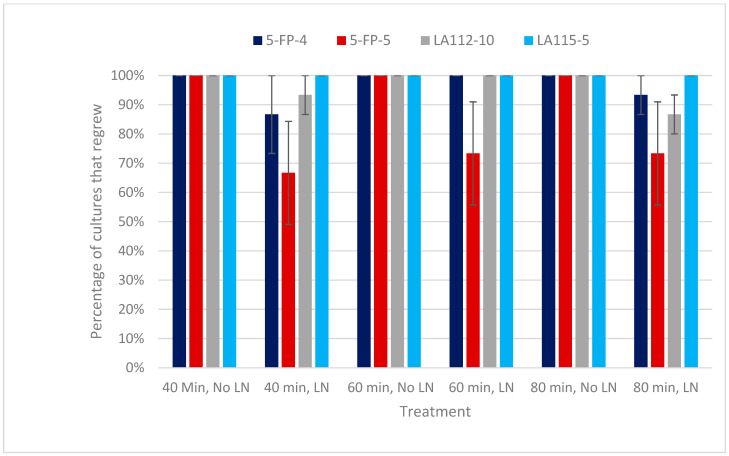
The effects of culture line (5-FP-4, 5-FP-5, LA112-10, and LA-115-5) and PVS2 exposure duration (40, 60, or 80 min) on percentage embryogenic tissue regrowth following cryostorage (LN) and in controls exposed to PVS2 without cryostorage, followed by transfer to semisolid IMM.. Combined data from all three vitrification experiments are shown. Data from each experiment were the percentage of the five cryovials of each culture line × treatment combination that regrew. Bars represent standard error. Percentage data shown here for each genotype × duration combination were not statistically analyzed. Results of GLM analysis of main (PVS2 exposure duration, genotype) effects and interaction effects are presented in the text.

**Figure 3 plants-13-00352-f003:**
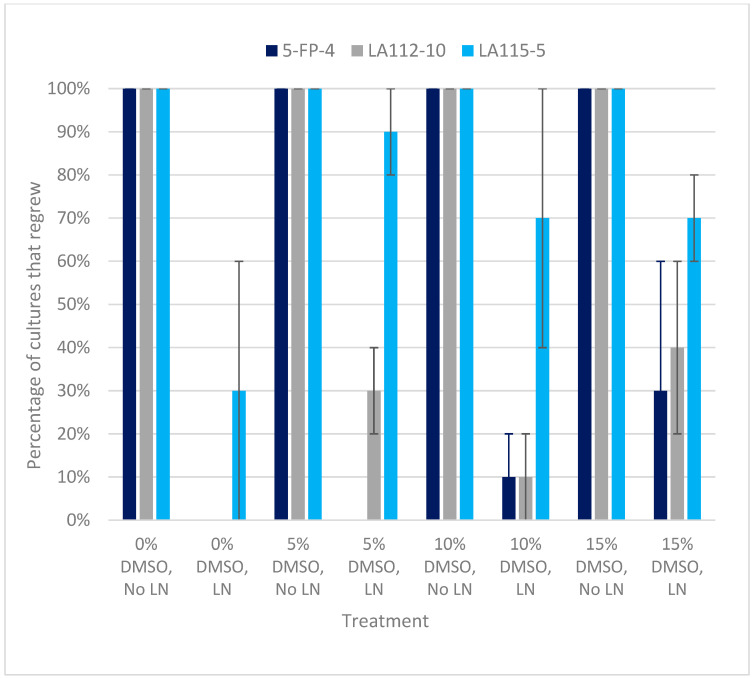
The effects of culture line (5-FP-4, LA112-10, and LA-115-5) and DMSO concentration (0%, 5%, 10%, and 15%) on percentage embryogenic tissue regrowth following cryostorage (LN) and in controls exposed to DMSO without cryostorage (No LN), followed by transfer to semisolid IMM. Combined data of two of the slow cooling experiments are shown. Data from each experiment were the percentage of five cryovials of each culture line × treatment combination that recovered. Bars represent standard error. Percentage data shown here for each genotype × DMSO concentration combination were not statistically analyzed. Results of the GLM analysis of the main (DMSO concentration and genotype) effects and interaction effects are presented in the text.

**Figure 4 plants-13-00352-f004:**
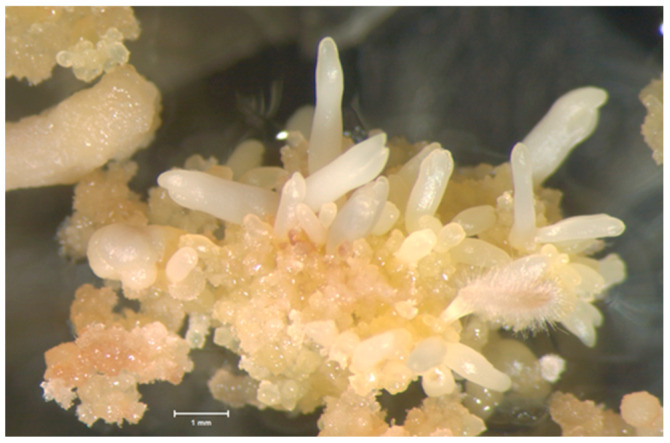
Somatic embryo production from culture line LA112-10 following recovery from cryostorage, regrowth on IMM, and transfer to EDM. Bar = 1 mm.

## Data Availability

Data are contained within the article.

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
