# Peer review of "Conservation of Green and White Ash Germplasm Using the Cryopreservation of Embryogenic Cultures"

_plants, 2024, doi:10.3390/plants13030352_

Round 1

Reviewer 1 Report

Comments and Suggestions for Authors

I have reviewed your manuscript “Conservation of green and white ash germplasm using cryopreservation of embryogenic cultures" submitted for publication in Plants. The manuscript falls under the scope of the journal Plants and is focused on the cryopreservation of embryogenic tissue in two ash species: green and white ash. Despite the fact that the procedures and cryoprotectants utilized in the research are not novel, this is the first attempt to cryopreserve embriogenic tissue of white and green ash, and the results obtained could be useful for the establishment of a cryobank of these two species in the future. I can't, however, recommend the text for publication at this time due to a number of concerns I have about various sections of it. If only the author(s) address carefully to all of my suggestions given in text itself, I could reconsider my opinion.

Due to the paper's structure and number of comments I have on various points in the manuscript, which makes it challenging to write a thorough and organized evaluation in this document, all comments are provided in the article itself (please refer to the attached file).

Reviewer 2 Report

Comments and Suggestions for Authors

See comments in the attached documents.

Verify the steps for the cryopreservation process conducted in this study.

Add more information in method section.

It would be good to show growth of somatic embryoids and plantlets development.

Reviewer 3 Report

Comments and Suggestions for Authors

All comments, observations and suggestions are made in the attached PDF file. Please check carefully all suggestions and specifically those related to 'replicates'.

Round 2

Reviewer 1 Report

Comments and Suggestions for Authors

The authors have incorporated all the corrections in the manuscript based on my previous comments. However, no data were provided on the somatic embryo or somatic seedling productivity of the regrown cultures, which could significantly enhance the quality of the work. Despite this, I recommend that the editor(s) accept the paper, as it represents the first report of a reliable cryopreservation protocol for green ash and white ash embryogenic tissues. In the attached document, I have outlined a few technical suggestions that should be considered when preparing the final version of the manuscript.

Reviewer 2 Report

Comments and Suggestions for Authors

N/A
